EMBO
Molecular Medicine

# Constitutive activation of DIA1 (DIAPH1) via C-terminal truncation causes human sensorineural hearing loss

Takehiko Ueyama[1,*], Yuzuru Ninoyu[1], Shin-ya Nishio[2], Takushi Miyoshi[3], Hiroko Torii[3], Koji Nishimura[3], Kazuma Sugahara[4], Hideaki Sakata[5], Dean Thumkeo[6], Hirofumi Sakaguchi[7], Naoki Watanabe[8,9], Shin-ichi Usami[2], Naoaki Saito[1] & Shin-ichiro Kitajiri[3,**]

## Abstract

*DIAPH1* encodes human DIA1, a formin protein that elongates unbranched actin. The c.3634+1G>T *DIAPH1* mutation causes autosomal dominant nonsyndromic sensorineural hearing loss, DFNA1, characterized by progressive deafness starting in childhood. The mutation occurs near the C-terminus of the diaphanous autoregulatory domain (DAD) of DIA1, which interacts with its N-terminal diaphanous inhibitory domain (DID), and may engender constitutive activation of DIA1. However, the underlying pathogenesis that causes DFNA1 is unclear. We describe a novel patient-derived *DIAPH1* mutation (c.3610C>T) in two unrelated families, which results in early termination prior to a basic amino acid motif (RRKR[1204–1207]) at the DAD C-terminus. The mutant DIA1(R1204X) disrupted the autoinhibitory DID-DAD interaction and was constitutively active. This unscheduled activity caused increased rates of directional actin polymerization movement and induced formation of elongated microvilli. Mice expressing FLAG-tagged DIA1(R1204X) experienced progressive deafness and hair cell loss at the basal turn and had various morphological abnormalities in stereocilia (short, fused, elongated, sparse). Thus, the basic region of the DAD mediates DIA1 autoinhibition; disruption of the DID-DAD interaction and consequent activation of DIA1(R1204X) causes DFNA1.

**Keywords** actin; deafness; DIAPH1; DFNA1; stereocilia
**Subject Categories** Genetics, Gene Therapy & Genetic Disease; Neuroscience

## Introduction

In the past two decades, extensive research on the genetics of nonsyndromic hereditary deafness (NSHD) has been conducted, leading to the discovery of about 100 genes essential for hearing (http://hereditaryhearingloss.org/): about 30 of these encode proteins that interact directly or indirectly with actin (Dror & Avraham, 2010; Drummond *et al*, 2012). Stereocilia, which are deflected by sound stimulation and are thus the key structures for hearing, are actin-based protrusions and exquisitely organized microvilli/filopodia composed of hundreds of parallel actin filaments with the plus ends at the distal tip; they are organized into precise rows of graded height (characteristic staircase patterns) on the apical surface of cochlear hair cells (HCs) (Frolenkov *et al*, 2004). Cochlear HCs are arranged in a single row of inner HCs (IHCs) and three rows of outer HCs (OHCs). Although IHCs and OHCs are believed to share common mechanotransduction machinery, they have distinct roles during sound detection: IHCs are true sensors, whereas OHCs function as amplifiers through an active process that involves stereociliary and somatic motility (Schwander *et al*, 2010). The high sensitivity of HCs depends on the coordinated/synchronized movement of stereocilia upon mechanical stimulation; thus, the precise organization of the length and shape of stereocilia, which is regulated by a large battery of genes, is indispensable.

Fifteen formin proteins, which nucleate and elongate unbranched/straight actin, are found in humans and can be classified into eight subfamilies (Campellone & Welch, 2010). The diaphanous-related formin (DRF) subfamily contains proteins commonly referred to as DIAs (also known as DIAPHs and mammalian homologs of *Drosophila* Diaphanous), which are the best-characterized

1 Laboratory of Molecular Pharmacology, Biosignal Research Center, Kobe University, Kobe, Japan
2 Department of Otorhinolaryngology, Shinshu University School of Medicine, Matsumoto, Japan
3 Department of Otolaryngology, Head and Neck Surgery, Graduate School of Medicine, Kyoto University, Kyoto, Japan
4 Department of Otolaryngology, Yamaguchi University Graduate School of Medicine, Ube, Japan
5 Kawagoe Otology Institute, Kawagoe, Japan
6 Medical Innovation Center, Kyoto University Graduate School of Medicine, Kyoto, Japan
7 Department of Otolaryngology-Head and Neck Surgery, Kyoto Prefectural University of Medicine, Kyoto, Japan
8 Department of Pharmacology, Kyoto University Graduate School of Medicine, Kyoto, Japan
9 Laboratory of Single-Molecule Cell Biology, Kyoto University Graduate School of Biostudies, Kyoto, Japan
*Corresponding author. Tel: +81 78 803 5962; Fax: +81 78 803 5971; E-mail: tueyama@kobe-u.ac.jp
**Corresponding author. Tel: +81 75 751 3346; Fax: +81 75 751 7225; E-mail: kitajiri@ent.kuhp.kyoto-u.ac.jp

 

class of formins. *DIAPH1* gene encodes one of three DIA isoforms, DIA1 (DIAPH1). A mutation in *DIAPH1* (c.3634+1G>T), in which the canonical splicing donor sequence is disrupted (AAGgtaagt becomes AAGttaagt), moves the splicing donor site to a cryptic site four base pairs (bp) downstream of the original site, resulting in a 4 bp (ttaa) insertion in the gene transcript. The insertion results in a frameshift near, but outside, the C-terminal end of the diaphanous autoregulatory domain (DAD) of DIA1, hereafter referred to as DIA1 (ttaa). The frameshift introduces 21 aberrant amino acids (aa) and removes 32 aa of wild-type (WT) DIA1 (p.Ala1212ValfsX22) (Fig 1A) and causes autosomal dominant nonsyndromic sensorineural hearing loss, DFNA1 (Lynch *et al*, 1997). DIAs have a modular domain organization consisting of the GTPase-binding domain (GBD), a partially overlapping N-terminal diaphanous inhibitory domain (DID), formin homology (FH) domains FH1 and FH2, and the C-terminal DAD (Fig 1A). Through an autoinhibitory intramolecular interaction between the DID and the DAD, which is regulated by Rho family GTPases (Amano *et al*, 2010), DIAs are held inactive in the resting state (Watanabe *et al*, 1999; Campellone & Welch, 2010; Kuhn & Geyer, 2014). DRFs including mouse Dia1 show directional movement over the distance of several micrometers in cells as they processively elongate actin filaments via their FH2 domain (Watanabe & Mitchison, 2002; Higashida *et al*, 2004). Given the location of the mutation, it has been speculated that disruption of the autoinhibitory intramolecular DID-DAD interaction, and consequent activation of DIA1, might underlie DFNA1 pathology. However, it is also possible that the DIA1(ttaa) mutation might compromise other functions of DIA1 that are not related to autoregulation. Thus, the underlying pathogenesis of DFNA1 remains to be determined at the molecular level.

Patients in the original DFNA1 pedigree demonstrate mild- and low-frequency deafness starting in childhood (about 10 years of

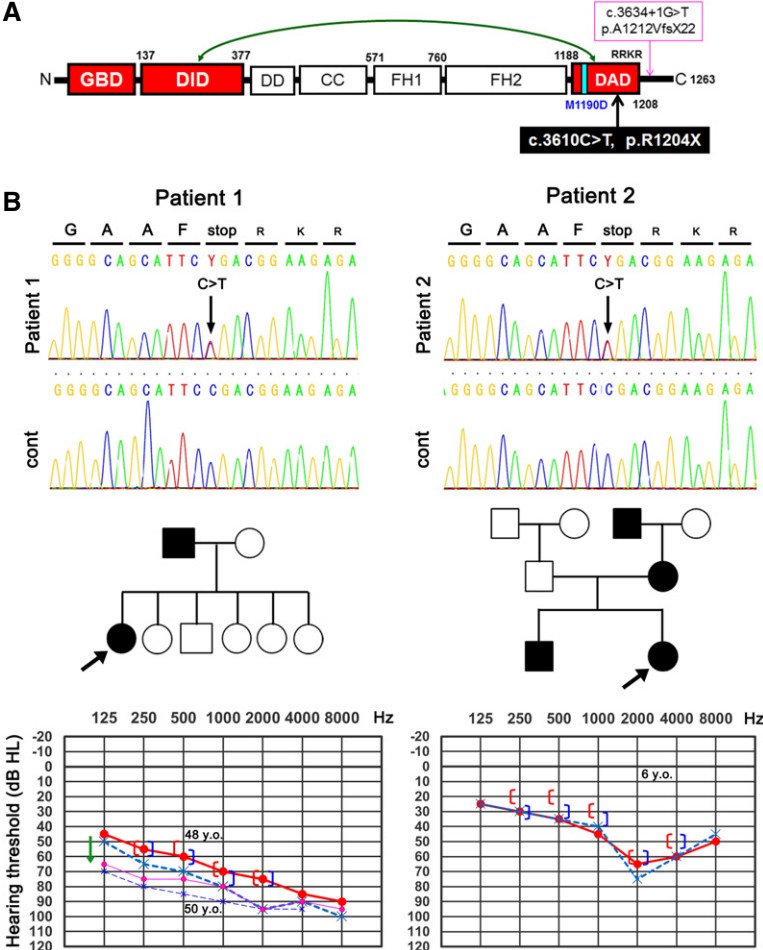

**Figure 1. Structural domains of DIA1 and patients with a novel DFNA1 mutation.**

A    The positions of M1190D, c.3610C>T (p.R1204X), and c.3634+1G>T (p.A1212VfsX22) with the latter being the only mutation previously associated with DFNA1. Bidirectional arrow shows the autoinhibitory intramolecular interaction between the diaphanous inhibitory domain (DID) and the diaphanous autoregulatory domain (DAD). GBD, GTPase-binding domain; FH1 and FH2, formin homology domain 1 and 2.

B    Electropherograms of DNA sequencing, family pedigrees, and audiograms of two unrelated patients. Electropherograms show a heterozygous c.3610C>T substitution resulting in a stop codon (R1204X). Arrows in family pedigrees indicate the patients identified in the present study. In audiograms, red/pink and blue indicate right and left, respectively; solid and dashed lines show air conduction hearing; and square brackets show bone conduction hearing. The bold line (at the age of 48 years) and thin line (at the age of 50 years) in the audiogram of patient 1 indicate progressive hearing loss (green arrow).

age), which develops into profound deafness involving all frequencies by adulthood (Lynch *et al*, 1997). Although DFNA1 was thought to originate due to sensorineural defects, evaluation of a young (8-year-old) DFNA1 patient with mild- and low-frequency deafness revealed a normal auditory brainstem response (ABR). It was therefore suggested that endolymphatic hydrops was involved in the early pathophysiology of the condition (Lalwani *et al*, 1998). Moreover, to date DFNA1 has only been reported in one Costa Rican family (called the "M" family because of the founder name, Monge). Thus, which of the two proposed underlying mechanisms is responsible for the development and clinical manifestations of DFNA1 remains unclear.

Here, we identified a novel *DIAPH1* point mutation (c.3610C>T), which results in early termination just before four sequential basic amino acids (RRKR$^{1204-1207}$) in the DAD, p.R1204X. This mutation (which we define as "R1204X") was found in two unrelated Japanese families and showed autosomal dominant progressive deafness in the high-frequency range that began in childhood. DIA1(R1204X) behaved as a constitutively active DIA1 mutant, as the mutation disrupted the autoinhibitory intramolecular DID-DAD interaction. This was associated with frequent directional actin polymerization movement and an increased number of elongated microvilli in cells. Furthermore, transgenic (TG) mice expressing FLAG-tagged DIA1(R1204X) showed progressive deafness that began in the high-frequency range in parallel with cochlear HC loss, predominantly in OHCs at the basal turn. Stereocilia were sparse and abnormally short, elongated, and fused. Comparison of *in vivo* models, including the novel and clinically relevant one we present here, will provide further insight into the molecular mechanisms that underlie the development of DFNA1 and the diverse clinical symptoms with which it is associated.

## Results

### Patients

By performing genomic sequencing, we identified a novel *DIAPH1* point mutation (c.3610C>T) generating the DIA1(R1204X) mutant, which has an early translation termination site just before four sequential basic amino acids (RRKR$^{1204-1207}$) in the C-terminal end of the DAD (Fig 1A and B). The location of the mutation is near the 4-bp (ttaa) insertion in *DIA1 (DIAPH1)* mRNA causing original DFNA1, but the ttaa insertion itself is located outside the DAD. This

mutation was found in two unrelated Japanese families. Family pedigrees of both patients showed an autosomal dominant inheritance of deafness (Fig 1B).

Patient 1 had a history of hearing loss that started in her childhood. She experienced slowly progressive deafness and was referred to an otologist at the age of 48. Magnetic resonance imaging of her brain, including the inner ear and VIII nerve, showed no abnormalities. Her audiograms (both air and bone conduction) showed bilateral deafness, predominantly in the high-frequency range, although all frequencies were affected (Fig 1B). Two years after her admission, her hearing deteriorated significantly, particularly in the low-frequency range (Fig 1B).

Patient 2 passed newborn hearing screening. She felt progressive deafness starting around 3 years old and was referred to an otologist at the age of 6. Her audiograms (both air and bone conduction) showed bilateral and high-tone deafness (Fig 1B).

### Elongation of microvilli by DIA1(R1204X) mutant

To examine the effect of DIA1(R1204X) on stereocilia, which are an exquisitely organized subtype of microvilli/filopodia, we studied the microvilli of MDCK cells cultured on membrane inserts. Because of the difficulties in observing and comparing the length of microvilli in reconstituted lateral images obtained by a confocal laser microscope, we established MDCK cells stably expressing mCherry-ESPIN1, which have elongated microvilli (about 2–3 μm) labeled by the mCherry red fluorescent protein. GFP-DIA1(M1190D), a constitutively active DIA1 mutant (Lammers *et al*, 2005), was localized to microvilli, induced microvilli elongation and altered cell shape (Fig 2). Unlike GFP-DIA1 (M1190D), the GFP-DIA1(R1204X) mutant was not discretely localized to microvilli, although it induced elongated and congested microvilli compared with surrounding nontransfected cells and WT GFP-DIA1-transfected cells (Fig 2). These results suggest that DIA1(R1204X) may be a constitutively active mutant of DIA1, albeit with weaker activity than DIA1(M1190D).

### Induction of elongated microvilli by DIA1(R1204X)

To confirm that DIA1(R1204X) is a mildly active DIA1 mutant, we utilized HeLa cells, which have previously been used in studies of DIA/Dia (Watanabe *et al*, 1999; Sakamoto *et al*, 2012). Although GFP-DIA1(M1190D) localized to the plasma membrane (PM) and

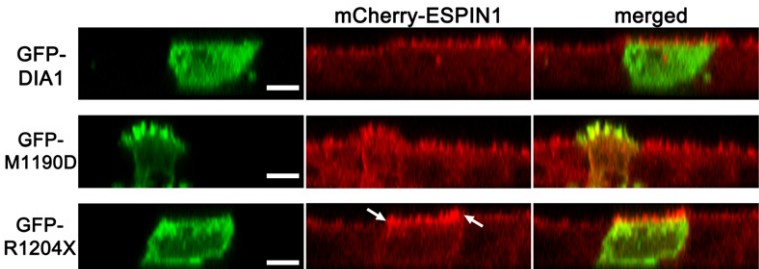

**Figure 2. DIA1(R1204X) induces elongation of microvilli in MDCK$^{mCherry-ESPIN1}$ cells.**
GFP-tagged DIA1, DIA1(M1190D), and DIA1(R1204X) plasmids were transfected into MDCK$^{mCherry-ESPIN1}$ cells cultured on membrane inserts. Twenty-four hours after transfection, a reconstructed lateral view of fixed cells was generated using a confocal laser fluorescence microscope. Arrows indicate the elongated microvilli visualized by mCherry-ESPIN1 with GFP-hDIA1(R1204X) expression. Scale bars: 5 μm. Representative of three experiments.

induced stress fiber (SF) formation, GFP-DIA1, GFP-DIA1(R1204X), or GFP-DIA1(ttaa) induced no remarkable SF formation (Fig 3A). However, GFP-DIA1(R1204X) had a dot-like localization adjacent to the PM (Fig 3A). Furthermore, 3D imaging revealed that GFP-DIA1 (R1204X) localized to microvilli, and induced elongated and thick microvilli on the surface (Fig 3B, Movie EV1). Nontransfected cells (Fig 3B, Movie EV1) or those expressing GFP-DIA1 or GFP-DIA1 (ttaa) (Movies EV2 and EV3) had no effect on microvilli phenotypes. Statistical analysis revealed that the longest microvilli in GFP-DIA1 (R1204X)-expressing cells were significantly longer than those in GFP-DIA1-expressing and nontransfected cells [R1204X ($n = 10$): $6.25 \pm 0.53$ µm, WT ($n = 10$): $3.09 \pm 0.23$ µm, nontransfected:

$3.12 \pm 0.22$ ($n = 10$); five independent experiments; $P < 0.0001$ (between WT and R1204X, nontransfected and R1204X) by one-way ANOVA followed by Bonferroni's *post hoc* test]. These results confirm that DIA1(R1204X) is a mildly active DIA1 that alters microvilli phenotypes, but that has no remarkable effects on SF formation.

### Disrupted intramolecular DID-DAD interaction in DIA1(R1204X)

To further confirm that DIA1(R1204X) is constitutively active due to the disrupted autoinhibitory interaction between the DID and the DAD, we performed pull-down assays using GST-tagged DID and biotin-tagged DADs: DAD(WT), DAD(R1204X), and DAD(M1190D)

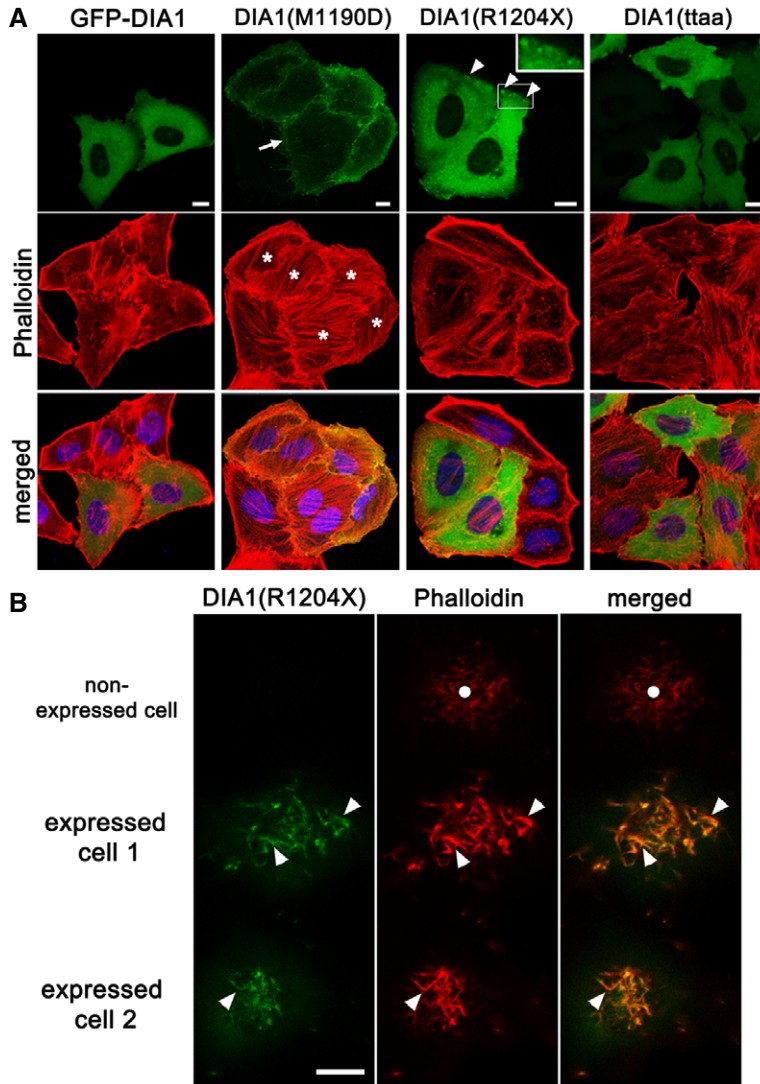

**Figure 3.  DIA1(R1204X) induces elongated and thick microvilli in HeLa cells.**
Various GFP-tagged DIA1 plasmids were transfected into HeLa cells and fixed 24 h after transfection. Fixed cells were stained with Alexa568-conjugated phalloidin and DAPI (blue) (A, B: representative of five experiments).

A   Stress fiber formation was observed with a confocal laser microscope. Note the induction of stress fibers by GFP-DIA1(M1190D) (asterisks). Arrow and arrowheads indicate a plasma membrane (PM) localization of GFP-DIA1(M1190D) and a dot-like localization of GFP-DIA1(R1204X) adjacent to the PM, respectively. Inset in the top panel of DIA1(R1204X) shows the magnified image of the region indicated by the rectangle. Scale bars: 10 µm.

B   Microvilli with or without GFP-DIA1(R1204X) expression were observed under a confocal laser microscope. Note elongated and thick microvilli (arrowheads) in GFP-DIA1 (R1204X)-transfected cells, but not in nontransfected cells (circles). 3D movies are available in Movie EV1 (R1204X), Movie EV2 (WT), and Movie EV3 (ttaa). Scale bar: 10 µm.

     

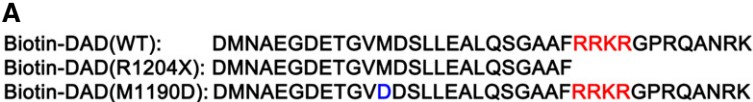

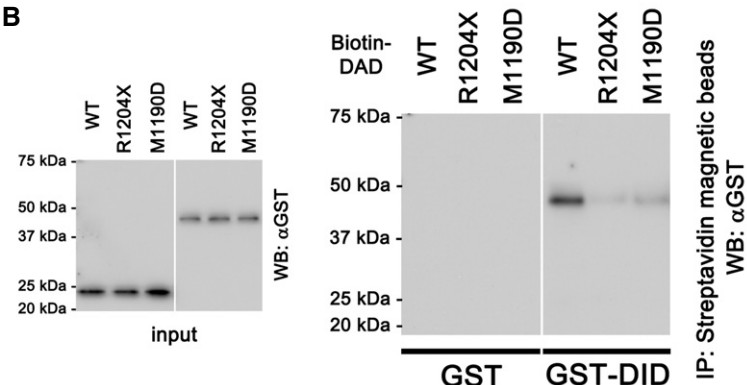

**Figure 4. R1204X mutation in the DAD disrupts the interaction between the DID and the DAD.**

A  Amino acid sequences of the biotin-labeled WT-, R1204X-, and M1190D-DAD. Underlined and red letters indicate the consensus motif (MDxLLxxL, essential for DID binding) and the unstructured basic region (enhances DID binding), respectively (Lammers *et al*, 2005).

B  Purified GST and GST-tagged DID proteins (50 nM) were mixed with biotin-labeled DADs (WT, R1204X, and M1190D; 50 nM) in binding buffer. After rotation, streptavidin-coupled magnetic beads were added to the solution, and the mixture was agitated. The material absorbed to the beads was eluted in Laemmli sample buffer and was subjected to SDS–PAGE, followed by immunoblotting using an HRP-conjugated GST Ab. Comparable levels of input (GST proteins) are confirmed in the left panel. Representative of five experiments.

Source data are available online for this figure.

(Fig 4A). The interaction between GST-DID and biotin-DAD(R1204X) was significantly weaker than that between GST-DID and biotin-DAD (WT), and similar to that between GST-DID and biotin-DAD(M1190D) (Fig 4B). These results suggest that the DIA1(R1204X) mutation disrupts the autoinhibitory intramolecular DID-DAD interaction, thereby rendering it constitutively active. However, DIA1(R1204X) was less active than DIA(M1190D) in cells, suggesting that factors other than the DID-DAD interaction may also regulate DIA1 activity.

### Importance of electrostatic interaction for the DID-DAD interaction

An acidic groove formed by acidic aa of the DID (E358, D361, E362, and D366 in mouse Dia1; E358, E361, E362, and D366 in human DIA1) has been identified by crystal structure analysis (Lammers *et al*, 2008; Nezami *et al*, 2010). Given that the DAD has four sequential basic aa ($RRKR^{1204–1207}$ in DIA1; RRKR is conserved in Dia1) at its C-terminus, electrostatic interactions may play an important role in the intramolecular DID-DAD interaction (Fig EV1A) (Lammers *et al*, 2008; Nezami *et al*, 2010; Kuhn & Geyer, 2014). Based on the prediction that exchange of $RRKR^{1204–1207}$ into EEEE would render DIA1 more active than DIA1(R1204X), we constructed a mutant DIA1($RRKR^{1204–1207}$/EEEEX) (Fig EV1A). GFP-DIA1 ($RRKR^{1204–1207}$/EEEEX) was localized to both the PM and microvilli, and its expression induced both microvilli elongation and SF formation (Fig EV1B, Movie EV4). These data suggest that this mutant is more active than DIA1(R1204X) and instead has similar activity to DIA1(M1190D) (Fig 3A).

To further confirm that electrostatic interactions play critical roles in the DID-DAD interaction, we made three additional

mutants: DIA1($RRKR^{1204–1207}$/EEEX), DIA1($RRKR^{1204–1207}$/EEX), and DIA1($RRKR^{1204–1207}$/EX) (Fig EV2A). PM localization was used as surrogate marker of activation, because active mutants, including DIA1(M1190D) and DIA1($RRKR^{1204–1207}$/EEEEX), are localized to the PM. DIA1($RRKR^{1204–1207}$/EEEX) and DIA1($RRKR^{1204–1207}$/EEX) were localized to the PM, whereas DIA1($RRKR^{1204–1207}$/EX) and DIA1(R1204X) showed a dot-like localization in the adjacent area to the PM. The degree to which each of the four mutants localized to the PM (or areas adjacent to the PM) depended on the number of acidic aa: DIA1($RRKR^{1204–1207}$/EEEEX) ≥ DIA1($RRKR^{1204–1207}$/EEEX) > DIA1($RRKR^{1204–1207}$/EEX) > DIA1($RRKR^{1204–1207}$/EX) > DIA1(R1204X) > WT DIA1 (Fig EV2B). These results indicate that electrostatic interactions play critical roles in the DID-DAD interaction, and that DIA1(R1204X) is a mildly active DIA1 mutant.

We made four additional mutants: DIA1($RRKR^{1204–1207}$/EEEE), DIA1($RRKR^{1204–1207}$/EEE), DIA1($RRKR^{1204–1207}$/EE), and DIA1($RRKR^{1204–1207}$/E) (Fig EV2C), which all retain the C-terminal 56 aa of DIA1 (1208–1263 aa). DIA1($RRKR^{1204–1207}$/E) was less strongly associated with areas adjacent to the PM when compared to DIA1($RRKR^{1204–1207}$/EX), with the degree of PM binding among this second set of mutants being DIA1($RRKR^{1204–1207}$/E) < Dia1 ($RRKR^{1204–1207}$/EE) < Dia1($RRKR^{1204–1207}$/EEE) < Dia1($RRKR^{1204–1207}$/EEEE) (Fig EV2D). Thus, we conclude that C-terminal aa (1208–1263 aa) are required to suppress DIA1 activation.

### Insufficient autoinhibition of DIA1(R1204X) in live cells

Previous fluorescence single-molecule speckle microscopy (SiMS) study revealed that the FH1-FH2 domain of mouse Dia1 shows processive movement in living cells based on its constitutive actin

polymerization activity (Higashida *et al*, 2004). In full-length WT Dia1, the polymerization activity of FH1-FH2 domain is suppressed by the autoinhibitory DID-DAD interaction, and binding of Rho family GTPases to the DID relieves this suppression (Watanabe *et al*, 1999). Compatible with this finding, WT Dia1 showed scarce processive movement in a SiMS study (Higashida *et al*, 2008), and microinjection of active RhoA(G14V) induced processive movement (Higashida *et al*, 2004).

We expressed GFP-tagged DIA1 molecules in living cells in order to evaluate the actin elongation activity of DIA1(R1204X) using fluorescence SiMS (Watanabe, 2012). GFP-DIA1(R1204X) expressed in XTC cells showed frequent processive movements (Fig 5B, Movie EV6) compared with GFP-DIA1(WT) (Fig 5A, Movie EV5), which indicated that the R1204X mutant had insufficient autoinhibition. GFP-DIA1(M1190D), a constitutively active mutant, also showed directional movements, which seemed more frequent than GFP-DIA1(R1204X) (Fig 5C, Movie EV7). To further quantify the differences between these three DIA1 variants, we counted the numbers of speckles that exhibited processive movement, and normalized these values by dividing them by the total fluorescent intensity of the observed area. This revealed significant differences in behavior, with the order (in terms of the most processive variant first) being GFP-DIA1(M1190D) > GFP-DIA1(R1204X) > GFP-DIA1 (WT) (Fig 5D). These data suggest that DIA1(R1204X) engenders

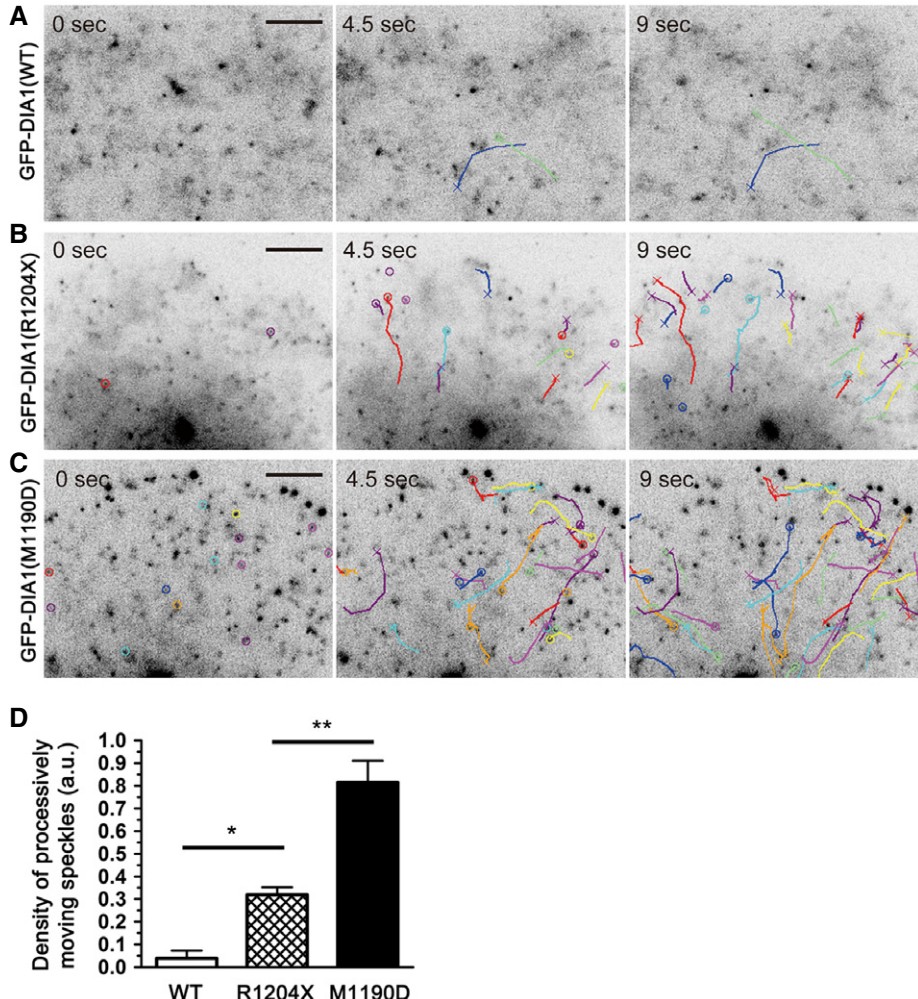

**Figure 5.  Single-molecule dynamics of the active mutant, hDIA1(R1204X), in XTC cells.**

A–D   GFP-tagged DIA1(WT), DIA1(R1204X), and DIA1(M1190D) were transfected into XTC cells. XTC cells were spread on the PLL-coated coverslips, and time-lapse images were acquired every 300 ms, illuminating the restricted areas near the cell edges, by using a single-molecule speckle microscope. Representative images are shown for DIA1(WT) (A, Movie EV5), DIA1(R1204X) (B, Movie EV6), and DIA1(M1190D) (C, Movie EV7). Circles indicate speckles showing directional movement over a distance of several micrometers. Trajectories of these speckles are also illustrated, and crosses were plotted at the end of the directional movement. Speckles showing directional movement were frequently observed in cells expressing GFP-DIA1(R1204X) (B), whereas such speckles were scarce in cells expressing GFP-DIA1 (WT) (A). However, the density of these speckles in GFP-DIA1(R1204X)-expressing cells appeared lower than that in GFP-DIA1(M1190D)-expressing cells, which is considered a fully active mutant (C). To evaluate the frequency of speckles indicating directional movement, we performed a quantitative analysis (D). The number of speckles showing directional movement during 3 continuous planes was counted and was normalized by the fluorescent intensity of each cell, which corresponds to the expression level of the GFP-tagged protein (mean $\pm$ SE). \*$P = 0.0288$ and \*\*$P = 0.0002$ by Bonferroni's *post hoc* test following one-way ANOVA (WT, $n = 5$; R1204X, $n = 6$; M1190D, $n = 6$). Time is in seconds. Scale bars: 5 µm.

upregulated actin elongation activity by disrupting the DID-DAD interaction, and that DIA1(R1204X) is less active than GFP-DIA1 (M1190D).

### DIA1(R1204X)-TG mice show progressive deafness and HC loss

To date, the possibility that DFNA1 caused by DIA1(ttaa) may be due to loss-of-function has not been completely excluded. Moreover, to the best of our knowledge, cochlear function in Dia1-KO mice has not been experimentally evaluated in detail (Thumkeo *et al*, 2013). At 5 weeks of age, Dia1-KO mice showed no HC loss or morphological abnormalities of substructures in the cochlea, including the organ of Corti (OC), spiral ganglion, spiral ligament, and stria vascularis (Fig 6A and B). The hearing function of Dia1-KO mice evaluated by ABR was no different to that of control mice at 5 weeks of age (Fig 6C). These results further suggest that DFNA1 caused by the original DIA1(ttaa) mutant is unlikely to be due to loss-of-function, dominant negative effects, or haploinsufficiency.

Based on our finding that DIA1(R1204X) is a mildly active mutant, and that a loss-of-function effect is unlikely to be the cause of DFNA1 in the context of this variant, we decided to make TG mice expressing FLAG-tagged DIA1(R1204X) under control of the CAG promoter (Fig 7A). First, we used quantitative PCR (qPCR) to determine the copy number of the transgene, FLAG-DIA1(R1204X), integrated into the genome ($3.86 \pm 0.26$; $n = 7$). Expression of FLAG-DIA1(R1204X) in the inner ear was confirmed using a FLAG antibody (Ab) (Fig 7B). Finally, we assessed expression levels of FLAG-DIA1(R1204X) protein in the inner ear, thymus, and heart by immunoblotting using a Dia1/DIA1 Ab, whose immunogen is 100% conserved between mouse Dia1 and human DIA1. The ability of the Ab to recognize the various Dia proteins with equal efficiency was confirmed by our observation that this Ab gave similar signal intensities when incubated with equal amounts of GFP-Dia1 and GFP-DIA1 protein obtained from transfected HEK293 cells (Fig 7C). The ratio of endogenous Dia1 + transgenic DIA1(R1204X)/endogenous Dia1 was $2.07 \pm 0.33$ in the inner ear and varied in other organs (for example, $4.89 \pm 0.69$ in the thymus, Fig 7C).

To assess cochlear function, we examined ABR with broadband click (2–4 kHz) and 8–32 kHz pure tone-burst stimuli, using 5- to 25-week-old DIA1(R1204X)-TG mice and control mice. At 10 weeks, DIA1(R1204X)-TG mice showed elevated ABR thresholds at high-frequency sounds (24 kHz and 32 kHz) compared with control mice (Fig 8A). Hearing impairment of DIA1(R1204X)-TG mice progressively deteriorated and involved all frequencies, including

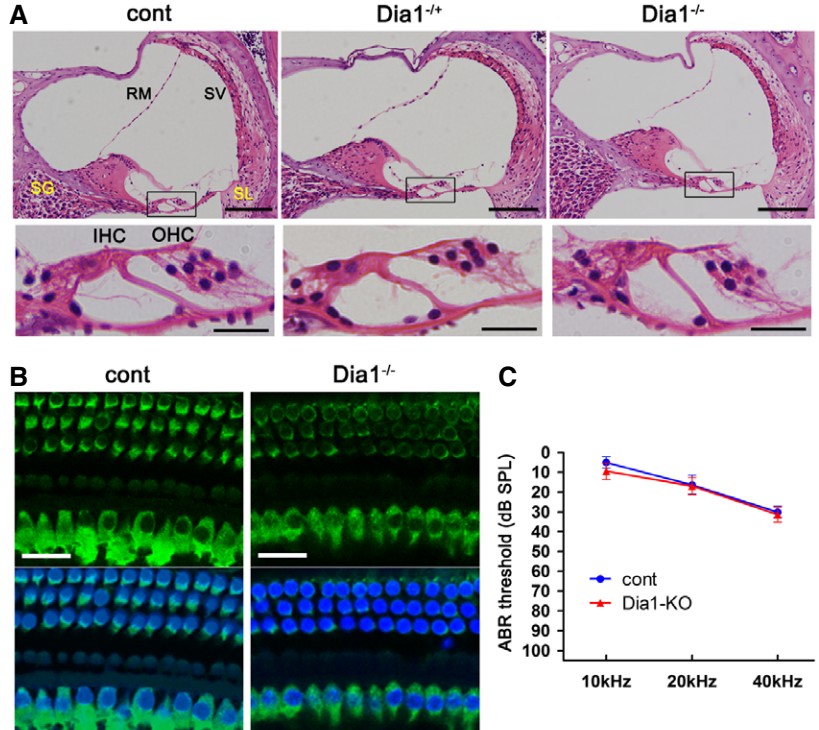

**Figure 6. No HC loss or hearing loss in Dia1-KO mice.**

Cochleae from control, heterozygous Dia1-KO (Dia1$^{-/+}$), and homozygous Dia1-KO (Dia1$^{-/-}$) mice were fixed at 5 weeks of age (A, B; representative of $n = 6$).

A   Paraffin-embedded 4-μm sections of cochlea were stained with hematoxylin and eosin. No hair cell (HC) loss (either IHC or OHC) is observed at the basal turn of the Dia1$^{-/+}$ or Dia1$^{-/-}$ cochleae. No obvious differences are observed in the spiral ganglion (SG), spiral ligament (SL), stria vascularis (SV), and Reissner's membrane (RM) in Dia1$^{-/+}$, or Dia1$^{-/-}$ mice compared with control mice. Lower panels (scale bars: 20 μm) are magnified images of the region indicated by the rectangle in upper panels (scale bars: 100 μm).

B   Basal turns of the dissected cochleae were immunostained using a Myo7a Ab followed by an Alexa488 secondary Ab with DAPI (blue) as a nuclear counterstain. No HC loss (either IHC or OHC) is observed in Dia1-KO (Dia1$^{-/-}$) mice. Scale bars: 20 μm.

C   ABR thresholds (10, 20, 40 kHz) in control and Dia1-KO (Dia1$^{-/-}$) mice at 5 weeks of age (dB SPL, mean $\pm$ SE). No hearing loss was observed in Dia1-KO (Dia1$^{-/-}$) mice ($n = 8$) compared with control mice ($n = 4$).

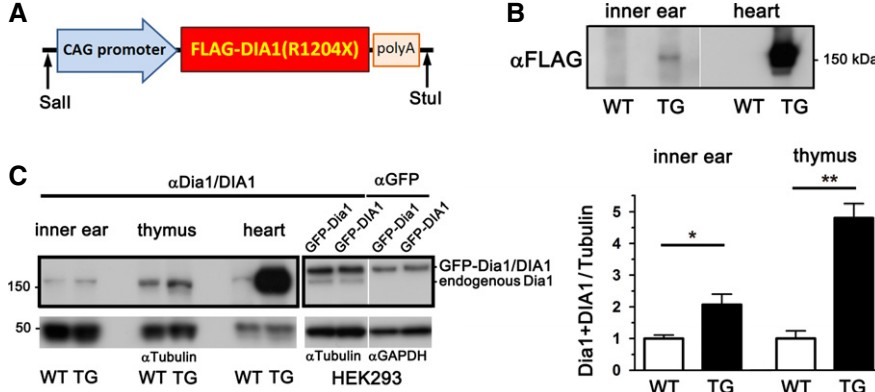

**Figure 7. Expression levels of the transgenic DIA1(R1204X) mutant in the inner ear of TG mice.**

A  Illustration of the linearized expression cassette containing the CAG promoter, 3xFLAG-tagged DIA1(R1204X), and SV40 poly(A) sequence used for injection into fertilized eggs.

B  Expression of FLAG-tagged DIA1(R1204X) in the inner ear and heart of TG, but not WT, mice is confirmed by immunoblotting using a FLAG Ab. Representative of three experiments.

C  Expression levels of FLAG-tagged DIA1(R1204X) relative to endogenous Dia1 in the inner ear, thymus, and heart were determined by immunoblotting using a Dia1/DIA1 Ab (inner ear, $n = 4$; thymus and heart, $n = 3$) and quantified in the inner ear and thymus (mean $\pm$ SE; *$P = 0.0226$ and **$P = 0.0016$ by two-tailed Student's $t$-test). Lysates of HEK293 cells overexpressing equal amounts of GFP-Dia1 and GFP-DIA1 (determined using a GFP Ab: right end) are detected with equal affinity by the Dia1/DIA1 antibody. Comparable loading of proteins is confirmed using a α-tubulin or GAPDH Ab.

Source data are available online for this figure.

low-frequency sounds at 25 weeks compared with control mice (Figs 8A and EV3A). Furthermore, we examined sex differences in ABR using 25-week-old mice: No significant difference was observed between male and female DIA1(R1204X)-TG mice (Fig EV3B).

Next, we examined the ultrastructure of the OC in DIA1 (R1204X)-TG mice using a scanning electron microscope (SEM). At 10 weeks, loss of OHCs started at the basal turn of DIA1(R1204X)-TG cochlea; in contrast, IHC was intact in all three turns (Fig 8B). At 25 weeks, loss of OHCs at basal and middle turns became significant, and loss of IHCs at three turns was also significant in DIA1 (R1204X)-TG mice (Fig 8C). Loss of OHC function in DIA1 (R1204X)-TG was physiologically confirmed using DPOAE (Fig 8D). High magnification at 25 weeks revealed that some OHCs had sparse and short stereocilia, and some IHCs had sparse and fused stereocilia (Fig 8E).

Finally, to confirm that progressive hearing loss and OHC- and basal turn-dominant HC loss was induced by the DIA1(R1204X) mutant, we bred the DIA1(R1204X) mice to homozygosity. When compared with heterozygous TG mice, the homozygotes showed exacerbated hearing impairment involving all frequencies (Fig EV4A) and exacerbated HC loss dominant in OHC and in the basal turn (Fig EV4B).

## Discussion

Here, we report a novel *DIAPH1* mutation that generates a mildly active DIA1(R1204X) mutant due to the disrupted autoinhibitory intramolecular DID-DAD interaction of DIA1; we also show that expression of this mutant *in vivo* is responsible for progressive deafness. The DIA1(R1204X) mutant has an early termination site immediately prior to a four amino acid basic sequence (RRKR[1204–1207]). The DAD of Dia1/DIA1 is composed of an amphipathic helix with

the central consensus motif MDxLLxxL followed by an unstructured basic region, including RRKR (Lammers *et al*, 2005). An acidic groove formed by acidic amino acids in the DID of Dia1/DIA1 is located adjacent to the MDxLLxxL recognition site (Lammers *et al*, 2008; Nezami *et al*, 2010). While the MDxLLxxL is essential for binding to a hydrophobic surface patch at the concave side of the DID, the basic region of the DAD is thought to interact with the acidic groove of the DID (Lammers *et al*, 2005; Wallar *et al*, 2006; Kuhn & Geyer, 2014). Thus, the basic region of the C-terminal end of the DAD is also an important contributor to the DID-DAD interaction. This would explain why mutation around RRKR[1204–1207] leads to constitutive DIA1 activation.

We previously reported inner ear HC-specific Cdc42-knockout (KO) mice, which show progressive deafness in parallel with HC loss with stereocilia degradation, including short, elongated, fused stereocilia (Ueyama *et al*, 2014). Furthermore, we found that the amount of active RhoA (GTP-form) is increased in Cdc42-knockdown (KD) cells (Ueyama *et al*, 2014). Because RhoA binds to and activates Dia1 (Watanabe *et al*, 1999), we suggested that the activated RhoA–Dia1 signaling axis is involved, at least to some extent, in HC loss and stereocilia degradation in Cdc42-KO mice (Ueyama *et al*, 2014). In support of our hypothesis, the degenerative stereocilia phenotypes we observed in TG mice expressing the active DIA1 mutant, DIA1(R1204X), are similar to those observed in Cdc42-KO mice. In addition to the absence of a hearing phenotype in Dia1-KO mice in the present study, patients with loss of functional variants in DIA1 (p.Q778X, p.F923 fs, and p.R1049X) suffer from microcephalus, blindness, and seizure, but are not deaf (Al-Maawali *et al*, 2016; Ercan-Sencicek *et al*, 2015). Moreover, a transgenic *Drosophila melanogaster* strain with only one diaphanous gene and expression of a constitutive active form of diaphanous protein by removing both the N-terminal GBD and the C-terminal DAD showed impaired response to sound (Schoen *et al*, 2010). Taken together,

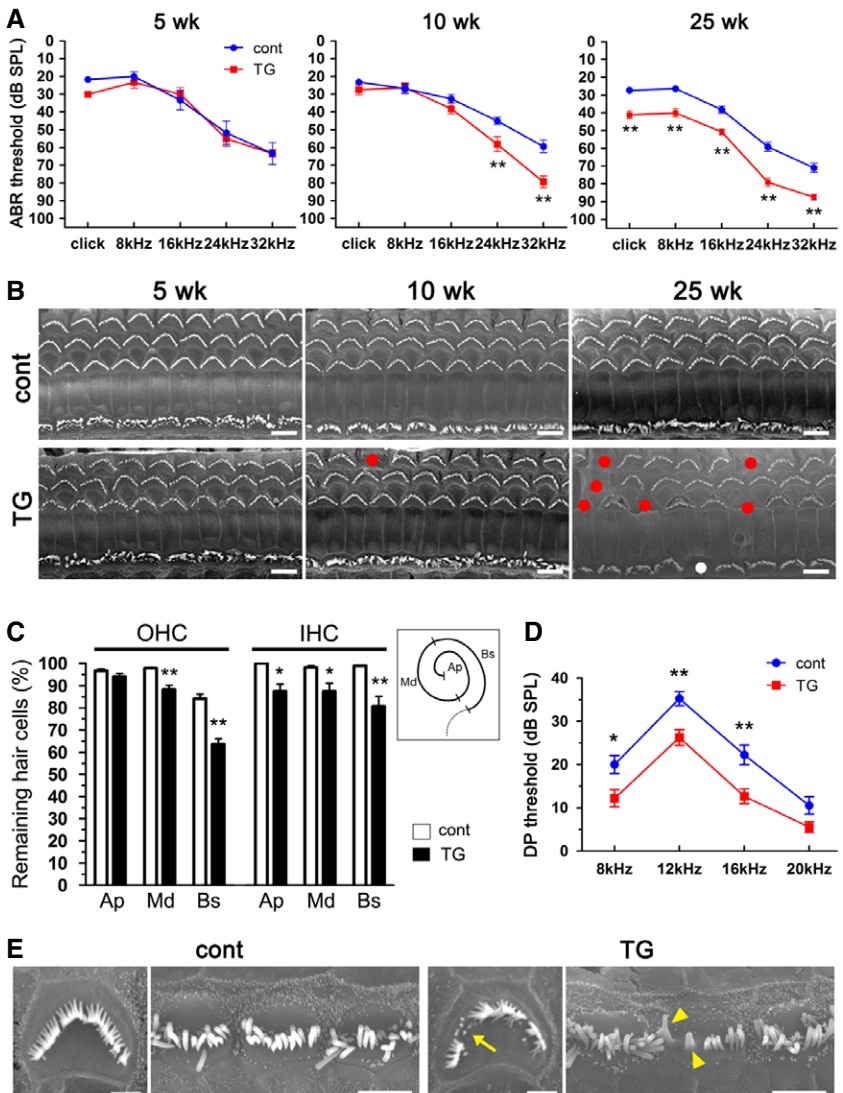

**Figure 8.  DIA1(R1204X)-TG mice show progressive hearing loss and a reduction of HC.**

A   Age-related click and pure tone-burst (8, 16, 24, 32 kHz) ABR thresholds (dB SPL, mean ± SE) in DIA1(R1204X)-TG (5 weeks, *n* = 6; 10 weeks, *n* = 16; 25 weeks, *n* = 42) and control mice (5 weeks, *n* = 6; 10 weeks, *n* = 16; 25 weeks, *n* = 33). Note the progressive hearing loss of DIA1(R1204X)-TG mice starting at high-frequency sounds from 10 weeks. **$P$ = 0.0039 (24 kHz) and **$P$ < 0.0001 (32 kHz) at 10 weeks; **$P$ < 0.0001 (click), **$P$ < 0.0001 (8 kHz), **$P$ = 0.0001 (16 kHz), **$P$ < 0.0001 (24 kHz), and **$P$ < 0.0001 (32 kHz) at 25 weeks by Bonferroni's *post hoc* test following two-way ANOVA.
B   Representative SEM images of the organ of Corti at the basal turn obtained from DIA1(R1204X)-TG and control mice at 5, 10, and 25 weeks of age. White and red circles show IHC and OHC losses, respectively. Note OHC-dominant HC loss. Scale bars: 5 μm.
C   The percentages of remaining OHC and IHC in each turn in DIA1(R1204X)-TG and control mice at 25 weeks of age (mean ± SE; *n* = 5). Schematic illustration shows the regions of the cochlea divided into three parts: Ap, apical turn; Md, middle turn; Bs, basal turn. **$P$ = 0.0017 (OHC at Md), **$P$ < 0.0001 (OHC at Bs), *$P$ = 0.0117 (IHC at Ap), *$P$ = 0.0330 (IHC at Md), and **$P$ = 0.0003 (IHC at Bs) by Bonferroni's *post hoc* test following two-way ANOVA.
D   DPOAE (f2 frequency at 8, 12, 16, 20 kHz) amplitude (dB SPL, mean ± SE) in DIA1(R1204X)-TG (*n* = 37) and control mice (*n* = 30) at 25 weeks of age. *$P$ = 0.0127 (8 kHz), **$P$ = 0.0029 (at 12 kHz), and **$P$ = 0.0013 (at 16 kHz) by Bonferroni's *post hoc* test following two-way ANOVA.
E   High magnification SEM images of OHCs and IHCs at the basal turn of cochlea obtained from DIA1(R1204X)-TG and control mice at 25 weeks of age. Note the sparse and short (arrow) stereocilia of OHC, and sparse and fused (arrowheads) stereocilia of IHC in DIA1(R1204X)-TG mice. Scale bars: 1 μm in OHCs and 5 μm in IHCs.

these data indicate that the pathogenesis of DFNA1 caused by *DIAPH1* mutations is likely due to gain-of-function rather than loss-of-function.

DIA1(R1204X)-TG mice showed progressive deafness (mainly in the high-frequency range in the early stage, but later progressing to the low-frequency range) in parallel with OHC-dominant HC loss. Various types of stereocilia degradation (Figs 8E and EV5),

including short, sparse (reduced number per HC), fused, and elongated stereocilia, were present; however, the elongated stereocilia phenotype was less frequent than the occurrence of short stereocilia. Nonsyndromic auditory neuropathy autosomal dominant 1 (AUNA1) is due to a mutation in the promoter region of *DIAPH3* (encodes DIA2), which increases the expression of both DIAPH3 mRNA and protein. The condition leads to progressive hearing loss

beginning in the second decade of life (Schoen *et al*, 2010). In a mouse model of AUNA1, in which TG mice overexpress WT *Diaph3* (encodes Dia2), animals experience progressive deafness and elongated and fused stereocilia selectively in IHCs without HC loss (Schoen *et al*, 2013). The different phenotypes between DFNA1 and AUNA1 both in human patients and mouse models are a strong indication that DIAPH1 (DIA1) and DIAPH3 (DIA2) have distinct expression patterns and/or function in HCs. Although accelerated actin polymerization due to Dia2 overexpression might explain the elongated stereocilia, the molecular pathology that leads to the predominance of short and sparse stereocilia in DIA1(R1204X)-TG mice is unclear. One possibility is that DIA1(R1204X) enhances actin polymerization not only at stereocilia but also at other sites. This could include adherens junctions (AJs), which are one of the sites where Dia1 is active (Sahai & Marshall, 2002), and would lead to misregulated/unbalanced actin turnover in cells. Impairment specifically at the apical surface of the neuroepithelium was reported in Dia1/Dia3-DKO mice, suggesting the primary functional site of Dia1 is at this site (Thumkeo *et al*, 2011). Transient overexpression of DIA1 (R1204X) in MDCK and HeLa cells induced elongated microvilli; however, long-term expression of the active mutant may induce misregulated/unbalanced actin polymerization not only at stereocilia but also other sites including AJs. This would ultimately lead to the phenotypic variation from enhanced (elongated stereocilia) to reduced (short and defective stereocilia) actin polymerization. Indeed, we observed the deformed cell–cell junction between HC and supporting cell (SC) in DIA1(R1204X)-TG mice (Fig EV5).

We obtained only one line of DIA1(R1204X)-TG mice among a total 41 founder (F0) lines, despite injecting the transgene on three separate occasions into a total of 412 fertilized eggs. qPCR analysis revealed that four transgene copies were integrated into the genome in heterozygous TG mice, which is significantly lower than the usual number (Brinster *et al*, 1981). Expression levels of DIA1(R1204X) protein in heterozygous TG mice varied in a tissue- and organ-specific manner; for example, levels were low in the inner ear and very high in the heart. Considering previous reports and our own experience, we suggest that high level DIA1(R1204X) expression in TG mice is cytotoxic and therefore leads to embryonic lethality. In support of this speculation, overexpression of active diaphanous protein in TG flies (Schoen *et al*, 2010) causes lethality in the pupal stage in some cases. Dia1 plays important roles in cell polarization, migration, axonogenesis, and exocrine vesicle secretion in the apical membrane (Kuhn & Geyer, 2014). Although mice with single KO of Dia3 show no apparent phenotype (Thumkeo *et al*, 2013), Dia2-KO leads to embryonic lethality (Watanabe *et al*, 2013). Very high levels of active DIA1 overexpression may also interfere with the function of Dia2 and Dia3. Furthermore, the HC loss and deafness phenotypes were exacerbated when heterozygous DIA1(R1204X)-TG mice were bred to homozygosity. From these observations, we infer that that phenotypes observed in DIA1(R1204X)-TG mice are unlikely to be due to integration site effects of the transgene, and that our DIA1(R1204X)-TG model faithfully recapitulates phenotypes of DFNA1 patients who inherited the condition in an autosomal dominant fashion.

The original report of DFNA1 was due to the DIA1(ttaa) mutation, which caused mild- and low-frequency deafness with normal ABR at the early stage of disease, and was diagnosed as endolymphatic hydrops (Lalwani *et al*, 1998). On the contrary, DFNA1 is associated with deafness (predominantly in the high-frequency range) in two patients caused by the newly described DIA1(R1204X) mutation, as well as in DIA1(R1204X)-TG mice. This suggests that the novel DFNA1 cases and the original DFNA1 are associated with distinct pathophysiologies. Moreover, patient 2 and DIA1(R1204X)-TG mice showed deafness beginning in the high-frequency range, and DIA1(R1204X)-TG mice progressed to deafness involving all frequencies. Patient 1 presented with high-frequency dominant hearing loss, but also loss of hearing at all other frequencies; this was probably since this particular clinical case was advanced, as the patient was 48 years old when diagnosed. Thus, the clinical manifestation of the novel DFNA1 subtype is likely deafness beginning with loss of the high-frequency ranges in childhood, which slowly progresses to deafness involving all frequencies. We showed that the C-terminal amino acid sequence after the DAD (1208–1263 aa) acts as a repressor of DIA1 activation (compare Fig EV2B and D). The DIA1(ttaa) mutant loses the basic amino acid sequence RK[1214–1215], which is the next basic amino acid cluster following RRKR[1204–1207], and has a shorter C-terminal (lacking 31 aa) than WT DIA1. This suggests that DIA1(ttaa) would be a weaker constitutively active DIA1 mutant than DIA1(R1204X). Although DIA1 (ttaa) elicited no apparent phenotype in the transient overexpression cell models (either MDCK or HeLa cells), long-term observation using animal models will facilitate elucidation of the mechanisms associated with DIA1(ttaa)-driven pathologies.

In summary, we identified a novel DFNA1 subtype caused by constitutively active DIA1 mutant, DIA1(R1204X), in two unrelated families, and confirmed its pathogenicity using cell and mouse models. Our DIA1(R1204X)-TG mice (in which expression levels of transgenic DIA1(R1204X) mutant protein in the inner ear of heterozygous mice are probably similar to DFNA1 patients) showed progressive hearing loss beginning in the high-frequency range in parallel with HC loss and stereocilia degradation that was dominant in OHCs and in the basal turn. Nonsense-mediated mRNA decay is unlikely to explain the pathogenicity of DFNA1 caused by DIA1(R1204X). Studies using knock-in mice of DIA1 (R1204X) may further confirm our results and will help to provide a molecular basis for the clinical diversity associated with DFNA1.

## Materials and Methods

### Human studies

*Subjects*
A total of 1,120 Japanese patients with bilateral nonsyndromic sensorineural hearing loss participated in this study. In total, 269 normal hearing controls, confirmed by pure tone audiometry, were also enrolled. Informed written consent was obtained from all subjects. This study was approved by the Shinshu University Ethical Committee, and the ethics committees of all other participating institutions listed in a previous report (Nishio & Usami, 2015).

*Genetic analysis*
We performed massively parallel DNA sequencing (MPS) analysis using an Ion Torrent Personal Genome Machine (PGM™) system

with the Ion AmpliSeq™ custom panel (Thermo Fisher) in 1,174 samples (905 hearing loss cases and 269 normal hearing controls) and using a HiSeq 2000 sequencer (Illumina) with the SureSelect target DNA enrichment kit (Agilent) in 215 cases.

### Amplicon library preparation and Ion PGM platform sequencing

Amplicon libraries of the target gene exons from 63 genes reported to cause nonsyndromic hearing loss were prepared with the Ion AmpliSeq™ panel. They were diluted to 20 pM, and the same amount of libraries from the six libraries of six patients was pooled for one sequence reaction. Emulsion polymerase chain reaction and sequencing were performed with the Ion PGM™ system using the Ion PGM™ 200 sequencing kit and the Ion 318™ chip (Thermo Fisher). A detailed protocol has been described elsewhere (Miyagawa *et al*, 2013; Nishio *et al*, 2015). The sequence data were mapped to the human genome sequence (build GRCh37/hg19), and the DNA variant regions were piled up with the Torrent Variant Caller plug-in software ver. 4.0 (Thermo Fisher).

### Targeted enrichment and HiSeq platform sequencing

The SureSelect target DNA enrichment kit, designed for the 112 potentially deaf-causing genes, was used in this study (Abe *et al*, 2003). The detailed gene list is described in our previous report (Miyagawa *et al*, 2013). A 3-μg DNA aliquot was fragmented using the Covaris™ S2 System (Covaris) into fragments of ~200-bp length. Furthermore, the target regions were enriched with a barcode adapter (Agilent). The same amount of libraries from each of 12 patients was pooled into one tube and analyzed in one lane of the HiSeq 2000 sequencer. The sequence data were processed by filtering the read quality QV = 30 as cutoff, and duplicate reads were also removed. After the filtering process, sequence reads were mapped to the human genome sequence (build GRCh37/hg19) using BWA software (Li & Durbin, 2009). After sequence mapping, the DNA variant regions were compiled with GATK software (Van der Auwera *et al*, 2013).

### Filtering detected variants

After their detection, the effects of the variants were analyzed using ANNOVAR software (Wang *et al*, 2010; Chang & Wang, 2012). We selected missense, nonsense, insertion/deletion, and splicing variants. Variants were further selected as < 1% of: (i) the 1000 Genomes Database (http://www.1000genomes.org/), (ii) the 6,500 exome variants (http://evs.gs.washington.edu/EVS/), (iii) the human genetic variation database (dataset for 1,208 Japanese exome variants) (http://www.genome.med.kyoto-u.ac.jp/SnpDB/index.html), (iv) the 269 in-house Japanese normal hearing controls, and (v) 1,000 control data in the deafness variation database (Shearer *et al*, 2014) (http://deafnessvariationdatabase.org).

### Animals

This study was approved by the Institutional Animal Care and Use Committees and carried out according to the Animal Experimentation Regulations of Kobe University (for DIA1-TG mice) and Kyoto University Graduate School of Medicine (for Dia1-KO mice). Animals were housed in specific pathogen-free conditions using the individually ventilated cage system (Techniplast, Japan) and

maintained on a 12-h light/dark cycle. Human *DIAPH1* with HindIII/NotI sites at the 5′ and 3′ ends was amplified by PCR and cloned into the HindIII/NotI site of p3xFLAG-CMV-10 (Sigma-Aldrich) using an In-Fusion HD Cloning kit (Takara Bio Inc.) and named 3xFLAG-DIA1. 3xFLAG-DIA1(R1204X) was made using QuickChange Lightning Site-Directed Mutagenesis kit (Agilent Technologies). Two PCR products, 3xFLAG-DIA1(R1204X) with a 5′ XhoI site and the SV40 polyadenylation signal (SV40 polyA) with StuI at the 3′ end, were amplified by PCR and cloned into the XhoI/StuI sites of the pCAGGS vector, which has the CAG promoter, using an In-Fusion HD Cloning kit (Fig 7A). After confirming the identity of the plasmid by sequencing and expression of 3xFLAG-DIA1 (R1204X) in HEK293 cells by immunoblotting, a purified fragment digested with SalI and StuI that contained the CAG promoter, 3xFLAG-DIA1(R1204X), and SV40 polyA was injected into a fertilized egg at the pronucleus stage obtained from C57BL/6 mice (Unitech Japan). Founder (F0) mice were screened by PCR using the following primer pair (named CAG/DIA1-1): 5′-CCTACAGCTCCTGGGCAACGTGTGCTGGTT-3′ (in the CAG promoter) and 5′-GTCGTTTAAGAATGTCCAATAAGGA-3′ (in DIA1: coding nucleotides 541–565), and positive mice were then screened to determine whether they possessed the whole sequence of the injected fragment using seven primer pairs, including CAG/DIA1-1, as listed in Table EV1. Offspring were genotyped by PCR using the CAG/DIA1-1 primer pair. F1 and later generations of heterozygous DIA1(R1204X)-TG mice, and F2 and later generations of homozygous DIA1(R1204X)-TG mice were used for subsequent analyses. Dia1-KO mice (having C57BL/6 genetic background) have been described previously (Sakata *et al*, 2007). Males were used in the analyses unless described (mice younger than 1 week were not differentiated based on sex). ABR measurement of DIA-TG mice was performed using both male and female and analyzed together at the age of 5, 10 and 25 weeks and also separately at the age of 25 weeks.

### Antibodies and chemicals

The following specific Abs were used (polyclonal unless indicated): monoclonal FLAG (clone M2, 1:1,000, Sigma-Aldrich), Myo7a (25-6790, 1:400, Proteus), and mDia1(aa 66–77) (DP4471, 1:500, ECM Biosciences), whose immunogen is 100% conserved between mouse Dia1 and human DIA1. HRP-conjugated Abs against GST (PM013-7, 1:5,000), α-tubulin (PM054-7, 1:2,000), GAPDH (M171-7, 1:2,000), and GFP (598–7, 1:2,000) were from MBL International. Alexa568-conjugated phalloidin (1:500) and Alexa488-conjugated secondary Abs (1:2,000) were from Invitrogen.

### Plasmids

Human *DIAPH1* was amplified by PCR and cloned into pEGFP (C1) (Invitrogen), to give a plasmid we named GFP-DIA1. A constitutively active mutant GFP-DIA1(M1190D) that confers the lowest binding affinity of the DAD to the DID (Lammers *et al*, 2005), GFP-DIA1(R1204X), which terminates at 1204 aa due to the 3610C>T mutation (GenBank/EMBL/DDBJ accession number: LC180357), and GFP-DIA1(ttaa), which is the original mutant associated with DFNA1, were introduced using a QuickChange kit (Stratagene). The DID of DIA1 (137–377 aa) was amplified by PCR and cloned into the BamHI/EcoRI sites of pGEX6P-1 and

named GST-DID. All plasmids were sequenced to confirm their identities.

## Cell culture

MDCK and HeLa cells (RIKEN BioResource Center) were grown in DMEM supplemented with 10% FBS (Nichirei), 100 units/ml penicillin, and 100 μg/ml streptomycin (Wako), in a 5% $CO_2$ humidified incubator at 37°C. MDCK cells with stable expression of mCherry-ESPIN1, which induces elongation of microvilli from the normal length of 500 nm to about 2–3 μm, were obtained by the selection in the presence of 0.4 mg/ml G418 (Wako) and named MDCK$^{mCherry-ESPIN1}$ cells. Elongated microvilli were visualized and phenotypically characterized with the aid of 3D reconstruction confocal laser microscopy. *Xenopus laevis* XTC cells (Watanabe, 2012) were grown at 21–23°C in 70% Leibovitz's L-15 medium (Invitrogen) with 10% FBS, without humidity control and $CO_2$ supply.

## Three-dimensional (3D) culture of MDCK cells and confocal fluorescence imaging

To grow microvilli on the apical surface, $5 \times 10^5$ MDCK$^{mCherry-ESPIN1}$ cells were grown on 0.45-μm polyester filter insert (12-mm-diameter Transwell, Corning) for 48 h to obtain complete confluency. Then, plasmids were transfected into cells using Lipofectamine LTX (Invitrogen). Twenty-four hours after transfection, cells were fixed with 4% paraformaldehyde (PFA) in 0.1 M PBS (pH 7.4). HeLa cells were seeded in 35-mm glass-bottomed dishes (MatTek), 48 h before transfection, and transfected using FuGENE 6. Thirty-two hours after transfection of plasmids, the cells were fixed using 4% PFA solution. After permeabilization with PBS containing 0.3% Triton X-100 (PBS-0.3T), fixed cells were stained with Alexa568-conjugated phalloidin with/without DAPI for 1 h at 23°C. Fluorescent images (2D and 3D) were obtained using a LSM700 confocal laser-scanning fluorescence microscope (Zeiss). For statistical analysis of the effect of DIA1 proteins on microvilli, the longest microvilli of HeLa cell from 3D reconstructed image were measured ($n = 10$ from five independent experiments).

## Live cell imaging and fluorescence single-molecule speckle microscopy (SiMS)

Single-molecule speckle microscopy (SiMS) imaging was performed as described previously (Watanabe, 2012). In brief, transfected *Xenopus laevis* XTC cells expressing GFP-tagged DIA1, DIA (R1204X), and DIA(M1190D) were allowed to spread on a PLL-coated glass coverslip attached to a flow cell in 70% Leibovitz's L15 medium without serum. The flow cell was placed on the stage of a microscope (BX52; Olympus) equipped with 75-W Xenon illumination and an EMCCD camera (Cascade II:512; Roper Scientific). Time-lapse imaging was performed at 21–23°C using the Metamorph software (Molecular Devices) every 300 ms. Fluorescent speckle microscopy was performed by observing cells expressing a low amount of GFP-tagged proteins using a PlanApo 100× NA 1.40 oil objective (Olympus), illuminating restricted areas near the cell edges. The number of speckles showing directional movement was counted for each cell. To roughly normalize these numbers in terms of expression level, they were divided by the values of

$S \times (F_{cell} - F_{bg})$. S is the dimension of the observed cell area (μm$^2$). $F_{cell}$ and $F_{bg}$ are the fluorescence intensities of cell edge and background, respectively.

## *In vitro* binding (pull-down) assays

Purified GST and GST-DID were obtained as described previously (Ueyama *et al*, 2007). Biotin-labeled DAD of DIA1: DAD(WT, 1179–1215 aa), DAD(R1204X), and DAD(M1190D), was synthesized by MBL international. GST-DID was mixed with biotin-DAD in 300 μl binding buffer (50 nM each) (Ueyama *et al*, 2007). After rotation for 2 h at 4°C, 40 μl streptavidin-coupled magnetic beads (Dynabeads M-280 Streptavidin; Invitrogen) was added to the solution, and the mixture was agitated for 90 min at 4°C. The precipitates were washed three times using a magnetic rack, and then, the material absorbed to the beads was eluted in Laemml sample buffer; the magnetic beads were then removed using a magnetic rack. The eluents were subjected to SDS–PAGE, followed by immunoblotting using an HRP-conjugated polyclonal Ab against GST. Bound antibodies were detected using the ECL detection system (Wako).

## Histocytochemistry

Dissected cochlear tissues were fixed by 4% PFA in 0.1 M PB (pH 7.4). After permeabilization with PBS containing 0.3% Triton X-100 (PBS-0.3T), fixed tissues were incubated with primary Ab for 2 h at 23°C in PBS-0.03T and 0.5% fat-free bovine serum albumin (BSA), followed by Alexa488-conjugated secondary Ab for 2 h at 23°C, and mounted in Prolong anti-fade (Invitrogen) with a coverslip. Immunostainings were observed under a LSM700 confocal microscope (Carl Zeiss). In the case of samples embedded in paraffin, after deparaffinization, HE staining was performed using hematoxylin solution and eosin solution (Muto Pure Chemicals). Samples were imaged under a light microscope (BX50; Olympus) with a DP26 camera (Olympus).

## Immunoblotting

Dissected inner ear tissues, thymus, and heart (from P3 mice, 2–3 mice in case of the inner ear) were lysed in homogenizing buffer (Ueyama *et al*, 2001) by sonication in the presence of protease inhibitor cocktail (Nacalai Tesque) and 1% Triton X-100. Total cell lysates were centrifuged at $12,000 \times g$ for 20 min at 4°C, and the supernatants were subjected to SDS–PAGE followed by immunoblotting for 2 h at 23°C using primary Ab diluted in PBS-0.03T containing 0.5% fat-free BSA. The bound primary Abs were detected with secondary Ab-HRP conjugates using the ECL detection system. For quantification, the relative expression levels of Dia1 and Dia1 + DIA1, which were detected using a mDia1(aa 66–77) Ab (DP4471, 1:500, ECM Biosciences), were normalized to that of α-tubulin.

## ABR and DPOAE measurements

To assess hearing, auditory brainstem response (ABR) and distortion product otoacoustic emission (DPOAE) were measured under anesthetization with chloral hydrate (500 mg/kg i.p., Nacalai Tesque) and on a heating pad. Blinded data analysis was performed by three otologists.

ABR was measured in DIA1(R1204X)-TG mice and Dia1-KO mice with their littermate control mice at the ages of 5, 10, and 25 weeks, as described previously (Kitajiri *et al*, 2004; Ueyama *et al*, 2014). ABR waveforms using sound stimuli of clicks or torn bursts at 8 kHz, 16, 24, or 32 kHz (10, 20, or 40 kHz for Dia1-KO mice) were recorded for 12.8 ms at a sampling rate of 40,000 Hz by using 50–5,000 Hz band-pass filter settings, and ABR waveforms from 500 stimuli were averaged. ABR thresholds (dB SPL) were defined by decreasing the sound intensity by 10 dB steps until the lowest sound intensity level resulting in a recognizable ABR wave pattern (mainly judged by recognition of wave III) was reached.

DIA1(R1204X)-TG mice and littermate controls aged 25 weeks were measured DPOAE, as described previously (Ueyama *et al*, 2014). DPOAE at frequency of 2f1–f2 was elicited using two primary tone stimuli, f1 and f2, with sound pressure levels of 65 and 55 dB SPL, respectively, with f2/f1 = 1.20. DPOAE amplitudes (dB SPL) were measured at f2 frequencies of 8, 12, 16, 20 kHz and plotted after substitution with noise floor amplitude.

### qPCR

Mouse tail genomic DNA was extracted from using NucleoSpin Tissue (Macherey-Nagel). Reactions were performed with negative control (no template) in a reaction mixture containing 35 ng genomic DNA, SYBR Premix Ex Taq II (Takara), and gene-specific primers (*Dia1*; *DIA1(R1204X)*; and *gapdh*, in Table EV1). Reactions were run at 95°C for 5 min, followed by 45 cycles of 95°C for 5 s, 60°C for 30 s, and 72°C for 20 s, using LightCycler 480 System II (Roche Diagnostics). For absolute quantification, standard curves were constructed using DIA1(R1204X) in pEGFP(C1) and Dia1 in pEGFP(C1) (Higashida *et al*, 2008), and the copy number of FLAG-DIA1(R1204X) integrated into mouse genome (the ratio of FLAG-DIA1(R1204X)/Dia1) was calculated. The copy number of Dia1 verified by *gapdh* as an internal control gene using the $\Delta\Delta C_T$ method (Livak & Schmittgen, 2001). The specificity of the amplified fragment was demonstrated by the presence of melting curves with single peaks.

### SEM

Freshly dissected inner ear tissues were fixed in 2% PFA, 2.5% glutaraldehyde (GA) in 0.1 M PB. OC epithelia were dissected in the same buffer and postfixed with 1% OsO4 in $H_2O$ for 1 h. For SEM, tissues were dehydrated in an ethanol series, followed by isoamyl acetate, and dried in a freeze dryer EYELA FD-5N (Tokyo Rika-kikai). Dried samples were sputter-coated with gold (Ion Sputter E-1010, Hitachi) and then examined using a TM3030Plus scanning electron microscope (Hitachi High-Technologies) at 15 kV. For statistical analysis of HC loss, the cochlea was divided three portions: basal, middle, and apical turn, as illustrated in Fig 8C.

### Statistical analysis

All data are presented as the mean ± SE. Two groups were compared using unpaired two-tailed Student's *t*-test. For comparisons of more than two groups, one-way ANOVA or two-way ANOVA was performed and followed by Bonferroni's *post hoc* test of pairwise group differences. Statistical analyses were performed

**The paper explained**

**Problem**
About 100 genes essential for hearing have been identified in nonsyndromic hereditary deafness: About 30 of these genes encode proteins that interact directly or indirectly with actin. *DIAPH1* encodes DIA1 (DIAPH1), a formin protein that elongates unbranched straight actin and has essential roles in modeling/remodeling cytoskeletons. Autosomal dominant nonsyndromic sensorineural hearing loss, DFNA1, which is characterized by progressive deafness starting in childhood, is caused from the c.3634+1G>T mutation in *DIAPH1* (p.Ala1212ValfsX22). The mutation occurs near the C-terminus of the diaphanous autoregulatory domain (DAD), which interacts with its N-terminal diaphanous inhibitory domain (DID), and may engender constitutive activation of DIA1 by disruption of the autoinhibitory DID-DAD interaction. Although the first report demonstrating that mutation in *DIAPH1* was associated with DFNA1 was published in 1997, the molecular mechanism underlying DFNA1 development has remained unclear.

**Results**
We describe a novel patient-derived *DIAPH1* mutation (c.3610C>T, p.R1204X) in two unrelated families, which results in early termination prior to a basic amino acid motif (RRKR[1204–1207]) at the DAD C-terminus. The mutant DIA1(R1204X) disrupted the autoinhibitory DID-DAD interaction and was constitutively active. Mice expressing FLAG-tagged DIA1(R1204X) experienced progressive deafness and outer hair cell loss at the basal turn and had various morphological abnormalities in stereocilia (such as short, fused, elongated, and sparse), which are actin-based protrusions sensing sound stimuli.

**Impact**
The molecular mechanism underlying DFNA1 development has been unknown for about 20 years since the first report of DFNA1. By demonstrating that a novel mutation of DIA1 in patients and a mouse model shows progressive hearing loss beginning in the high-frequency range, the present study provides the novel concept that constitutive activation of DIA1 leads to DFNA1. Our work also provides a conceptual framework upon which future investigations into the clinical diversity associated with DFNA1 can be built.

using Prism 6.0 software (GraphPad); $P < 0.05$ (*$P < 0.05$ and **$P < 0.01$) was considered statistically significant.

**Expanded View** for this article is available online.

## Acknowledgements

We thank Tatsuya Uebi, Biosignal Research Center, Kobe University, for analysis of qPCR experiments. This work was supported by JSPS KAKENHI (to TU, S-iK, NS), the Takeda Science Foundation (to TU), the Uehara Foundation (to TU), the third step of Visionary Research Grant (a stepwise encouraging program) by Takeda Science Foundation (to S-iK).

## Author contributions

TU, S-iU, and S-iK planned the project. S-yN, KS, HSakata, S-iU, and S-iK performed human studies. TU, YN, HSakaguchi, and NS performed biochemical and cellular experiments and animal studies using DIA1(R1204X)-TG mice. TM, NW, and S-iK performed the SiMS experiments for analysis of actin polymerization. HT, KN, DT, and S-iK performed experiments using Dia1-KO mice. TU, HSakaguchi, and S-iK analyzed data, and TU wrote the manuscript.

## Conflict of interest

The authors declare that they have no conflict of interest.

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
