## [Review Process File · EMBO Molecular Medicine]

Constitutive activation of DIA1 (DIAPH1) via C-terminal truncation causes human sensorineural hearing loss

Takehiko Ueyama, Yuzuru Ninoyu, Shin-ya Nishio, Takushi Miyoshi, Hiroko Torii, Koji Nishimura, Kazuma Sugahara, Hideaki Sakata, Dean Thumkeo, Hirofumi Sakaguchi, Naoki Watanabe, Shin-ichi Usami, Naoaki Saito, Shin-ichiro Kitajiri

Corresponding author: Takehiko Ueyama, Biosignal Research Center, Kobe University

Review timeline:

Submission date:	17 May 2016
Editorial Decision:	09 June 2016
Revision received:	01 September 2016
Editorial Decision:	06 September 2016
Revision received:	13 September 2016
Accepted:	15 September 2016

Transaction Report:

Editor: Céline Carret

1st Editorial Decision

09 June 2016

Thank you for the submission of your manuscript to EMBO Molecular Medicine. We have now heard back from the two referees whom we asked to evaluate your manuscript. Although the referees find the study to be of potential interest, they also raise a number of concerns that need to be convincingly addressed in the next final version of this article.

You will see that both referees found the study of interest, although while ref1 is mainly supportive, requesting a couple of additional experiments, ref2 is more demanding. Nevertheless, we feel that both reports are very clear and should you perform the additional experiments and explain better your results and ameliorate the discussion, the study would be improved and overall more conclusive.

As such, given the balance of these evaluations, we feel that we can consider a revision of your manuscript if you can address the issues that have been raised within the space and time constraints outlined below. Please note that it is EMBO Molecular Medicine policy to allow only a single round of revision and that, as acceptance or rejection of the manuscript will depend on another round of review, your responses should be as complete as possible.

Revised manuscripts should be submitted within three months of a request for revision; they will otherwise be treated as new submissions, except under exceptional circumstances in which a short extension is obtained from the editor.

I look forward to seeing a revised form of your manuscript as soon as possible.

***** Reviewer's comments *****

Referee #1 (Remarks):

Overall, this is a high quality paper that makes valuable contributions to the understanding of hearing and deafness. In particular, mutations were identified in human Dia 1 by genomic analysis of patients of suffering from hearing loss. Through reasonable cellular and biochemical analysis, the authors determined that the mutations increased hDia1 activity and that this gain of function changes the regulation of the actin cytoskeleton. Considering the importance of actin regulation in stereocilia formation and maintenance, as well as the existing uncertainty over the function of Dia type proteins in sensory hair cells, this work is likely to be impactful. The results of this study are generally convincing but would ideally be strengthened by addressing the points listed below.

Specific concerns.

The level of overexpression in the transgenic model may be high resulting in non-specific effects. Its utility would be greatly enhanced if the authors could quantify the expression level either by qPCR or western blot of cochlear extracts. This would provide a more useful context for interpreting the phenotype. I agree with the authors that a knockin may eventually prove to be more informative, but is beyond the scope of this paper.

The cell biological assays appear to be well designed but without quantification. Perhaps the authors could measure some aspect of the phenotype, such as protrusion length, to better represent the cellular effect.

Referee #2 (Remarks):

page 2, Abstract (and throughout): "hDIA1" and "mDia1" are frequently used in the literature, abbreviations that use "h" for "human" and "m" for "mouse" are discouraged, although inconsistently. DIA1 is the accepted protein abbreviation and the authors can use "human DIA1" when they need to be specific.

page 3, Introduction: delete "is" i.e. "(DRF) subfamily is contains proteins"

page3, (and Figure 1): The description of the original DFNA1 mutation, described by Lynch et al. 1997, is misleading. The original mutation is a single nucleotided replacement, designated IVS17DS, G-T, +1 in ClinVar although it actually occurs in intron 27 (not 17). This removes a donor splice site, and a cryptic splice site 4 bps downstream is used. Thus, TTAA is inserted in the message - not in the germline DNA, as implied by the c.3634_3635insTTAA nomenclature.

page 8-9: The assays (Fig. 5 and 6) introducing glutamate residues in the DAD seem tangential here. The conclusions derived from these experiments do not directly support, or contradict, the hypothesis that the R1204X mutation causes DFNA1 by creating a constitutively activated DIA1. Given that these experiments are essentially unnecessary, they occupy an inordinate amount of the paper, and would be more appropriate for the supplement.

page 10: Conversely, the phenotypic description of the Dia1-KO mouse is directly relevant to the main hypothesis of the article, and should be in the main paper not the supplement. The authors rightly point out that the possibility that the "TTAA" allele was a loss of function mutant has never been adequately addressed. By providing data on the KO phenotype, and comparing it to the R1204X phenotype, the authors are in a position to provide insight to this long standing question. Fig. EV3 should be in the regular paper, possibly replacing Figs. 5 and 6.

Also, Lynch et al. reported recovering mutant cDNA from lymphoblastoid cell lines, which argued against this frame-shifting mutation causing nonsense-mediated decay. But their assay was not quantitative, and was performed before numerous alternative splice forms of DIAPH1 had been described. The authors should consider the possibility that the p.R1204X mutation leads to nonsense-mediated decay, at least in the Discussion.

page 10: How do the authors account for the difference in behavior between the mouse and human forms of wild-type DIA1 in the actin elongation assay? Does this not call into question the conclusions of the assay? The authors also might consider in the Discussion the implications, given their choice of human DIA1 as the basis of the transgene insertion.

page 11, (and Figure 7): The ABR results depicted in Figure 7C do not correspond well with description in the Methods section. In the Methods it is stated that the ABR waveform was recorded from a 40 kHz stimulus (alone?). Figure 7C shows results from clicks and pure-tones up to 32 kHz. The y-axis in 7C is labelled as dB, while in the Methods it is indicated that the units are dB SPL. According to the Methods the pure-tone stimuli were altered in 10 dB steps. Given that paradigm it is surprising that a statistically significant difference of 10 dB could be found between controls and TG mice at multiple stimulus levels. The error bars on the Figure are not indicated in the legend - are they stdev or SE? In either case they are surprisingly small given 10 dB steps and the small numbers of tested animals. When attempting to demonstrate subtle threshold differences it is important that the ABR scoring be blinded (was it?) and that example traces from the two genotypes be shown. It is also important that the data for males and females be presented independently - the authors state that there was no difference but they should show the data. On a C57Bl6 background there are often greater than 5-10 dB threshold differences between male and female WT littermates, although more reliably at ages older than 25 weeks.

The authors show representative SEM images comparing the cochleae of control and TG animals in 7D, and the OHC loss depicted for the TG genotype is provocative. However, OHC on a C57Bl6 background is common and dependant on age and position in the cochlea. Could the authors provide more quantitative data? e.g. Hair cell counts per length-window at base, middle and apex? Given the subtle ABR differences it would be surprising that there would be a pronounced histological difference that need not be quantified. Finally, if the main difference is in OHC numbers could the authors show DPOAE results as well?

1st Revision - authors' response

01 September 2016

Response to Reviewer #1

1. The level of overexpression in the transgenic model may be high resulting in non-specific effects. Its utility would be greatly enhanced if the authors could quantify the expression level either by qPCR or western blot of cochlear extracts. This would provide a more useful context for interpreting the phenotype.

In accordance with the thoughtful reviewer's suggestion, we examined the expression levels of the mutant DIA1(R1204X) in the inner ear, thymus, heart by western blot using a Dia1/DIA1 antibody, which recognizes mouse Dia1 and human DIA1 with equal efficiency. Using ImageJ software we determined that the ratio of Dia1+ DIA1(R1204X)/Dia1 in the inner ear from four independent experiments was 2.07 ± 0.33 . Representative immunoblots and statistical analysis are shown in a new Figure 7C. We believe that this new information supports usefulness of our TG mice as a model of DFNA1, which shows that autosomal dominant inheritance is due to the DIA1(1204X) mutant.

2. The cell biological assays appear to be well designed but without quantification. Perhaps the authors could measure some aspect of the phenotype, such as protrusion length, to better represent the cellular effect.

As suggested, we quantified the maximum length of microvilli in HeLa cells which were transfected WT GFP-DIA1, GFP-DIA1(R1204X): 6.25 ± 0.53 mm in GFP-DIA1(R1204X) expressing cells ($n = 10$), 3.09 ± 0.23 mm in GFP-DIA1 expressing cells ($n = 10$), and 3.12 ± 0.22 mm in non-transfected cells ($n = 10$); $P < 0.0001$ (between WT and R1204X, non-transfected and R1204X) by one-way ANOVA followed by Bonferroni's *post hoc* test). This result is described in the main text on page 7. In addition, we present SiMS data with quantification showing actin

elongation activity as a new Figure 5, (this was the old Fig. EV2, which lacked quantification and DIA1(M1190D) data). We believe these quantitative data more clearly demonstrate the enhanced activity of DIA1(R1204X) for actin elongation/polymerization.

Response to Reviewer #2

1. page 2, Abstract (and throughout): "hDIA1" and "mDia1" are frequently used in the literature, abbreviations that use "h" for "human" and "m" for "mouse" are discouraged, although inconsistently. DIA1 is the accepted protein abbreviation and the authors can use "human DIA1" when they need to be specific.

In accordance with the reviewer, we changed hDIA1 and mDia1 into DIA1 and Dia1, respectively, and use human DIA1 and mouse Dia1 when species identity is critical.

2. page 3, Introduction: delete "is" i.e. "(DRF) subfamily is contains proteins"

Thank you for pointing out this error. We deleted "is".

3. page3, (and Figure 1): The description of the original DFNA1 mutation, described by Lynch et al. 1997, is misleading. The original mutation is a single nucleotide replacement, designated IVS17DS, G>T, +1 in ClinVar although it actually occurs in intron 27 (not 17). This removes a donor splice site, and a cryptic splice site 4 bps downstream is used. Thus, TTAA is inserted in the message - not in the germline DNA, as implied by the c.3634_3635insTTAA nomenclature.

Thank you for pointing out this error and advising to prevent misleading the readers about the original DFNA1 mutation, described by Lynch. We changed "c.3634_3635insTTAA" into "c.3634+1G>T". We decided not to use of alternative nomenclature "IVS27DS, G>T, +1" to prevent confusion causing from the wrong nomenclature "IVS17DS, G>T, +1" which is widely used for describing the original DFNA1 mutation. We also changed the sentences describing the mechanism of ttaa insertion in DIA1 mRNA caused by the G>T mutation at the splicing donor site in page 3 as follows: "A mutation in DIAPH1 (c.3634+1G>T), in which the canonical splicing donor sequence is disrupted (AAGgtaagt becomes AAGttaagt), moves the splicing donor site to a cryptic site four base pairs (bp) downstream of the original site, resulting in a 4 bp (ttaa) insertion in the gene transcript." In addition, we changed the sentence in page 3 as follows: "The location of the mutation is near the 4 bp (ttaa) insertion in *DIA1* (*DIAPH1*) mRNA causing original DNFA1, but the ttaa insertion itself is located outside the DAD."

4. page 8-9: The assays (Fig. 5 and 6) introducing glutamate residues in the DAD seem tangential here. The conclusions derived from these experiments do not directly support, or contradict, the hypothesis that the R1204X mutation causes DFNA1 by creating a constitutively activated DIA1. Given that these experiments are essentially unnecessary, they occupy an inordinate amount of the paper, and would be more appropriate for the supplement.

As suggested by the reviewer, the original Figures 5 and 6 were moved to supplementary Figures (new Fig. EV1 and new Fig. EV2A, B).

5. page 10: Conversely, the phenotypic description of the *Dial*-KO mouse is directly relevant to the main hypothesis of the article, and should be in the main paper not the supplement. The authors rightly point out that the possibility that the "TTAA" allele was a loss of function mutant has never been adequately addressed. By providing data on the KO phenotype, and comparing it to the R1204X phenotype, the authors are in a position to provide insight to this long-standing question. Fig. EV3 should be in the regular paper, possibly replacing Figs. 5 and 6.

As suggested, we moved Figure EV3 into the regular paper as a new Figure 6. With this removal of the new Figure 5, we have replaced Figure 5A with new panels that clearly show that no abnormal morphological phenotype was observed in the cochlea of *Dial*^{-/+} or *Dial*-KO (*Dial*^{-/-}) mice compared with that of control mice.

6. Also, Lynch et al. reported recovering mutant cDNA from lymphoblastoid cell lines, which argued against this frame-shifting mutation causing nonsense-mediated decay. But their assay was not quantitative, and was performed before numerous alternative splice forms of *DIAPH1* had been described. The authors should consider the possibility that the p.R1204X mutation leads to nonsense-mediated decay, at least in the Discussion.

Thank you for the thoughtful suggestion about the possibility of nonsense-mediated mRNA decay of the *DIA1*(R1204X) mutant. In addition to the expression of FLAG-*DIA1*(R1204X) protein in the inner ear using a FLAG antibody (new Figure 7B), we evaluated levels of FLAG-*DIA1*(R1204X) using a *Dial*/*DIA1* antibody (as described in the response to the Reviewer 1), which recognizes mouse *Dial* and human *DIA1* with equal affinity, and showed the expression levels of *DIA1*(R1204X) in the inner ear (new Figure 7C). Judging from these results and the lack of phenotype in *Dial*-KO mice, we infer that nonsense-mediated mRNA decay of the *DIA1*(R1204X) transcript is unlikely to explain the pathogenicity. As suggested, this conclusion is described in the final section (in summary) of the Discussion: page 17.

7. page 10: How do the authors account for the difference in behavior between the mouse and human forms of wild-type *DIA1* in the actin elongation assay? Does this not call into question the conclusions of the assay? The authors also might consider in the Discussion the implications, given their choice of human *DIA1* as the basis of the transgene insertion.

We described that both WT mouse *Dial* and WT human *DIA1* show low activity for actin elongation in the resting states, and no functional difference between *Dial* and *DIA1* has been reported. The confusion of the reviewer may be from our incomplete sentences: "the FH1-FH2 domain of m*Dial* has processive actin elongation activity in living cells, and that this activity leads to rotation along the axis of actin filaments. The nucleation activity of m*Dial* is regulated by the autoinhibitory DID-DAD interaction, which is released by binding of Rho family GTPases to the DID. SiMS studies of WT m*Dial* revealed a low basal level of processive movement, whereas microinjection of active RhoA(G14V) promotes processive actin elongation." To clarify this misleading section, we re-wrote the sentences as follows: "the FH1-FH2 domain of mouse *Dial* shows processive movement in living cells based on its constitutive actin-polymerization activity. In full-length WT *Dial*, the polymerization activity of FH1-FH2 domain is suppressed by the autoinhibitory DID-DAD interaction, and binding of Rho family GTPases to the DID relieves this suppression. Compatible with this finding, WT *Dial* showed scarce processive movement in a SiMS study, and microinjection of active RhoA(G14V) induced processive movement, page 9-10.

8. page 11, (and Figure 7): The ABR results depicted in Figure 7C do not correspond well with description in the Methods section. In the Methods it is stated that the ABR waveform was recorded from a 40 kHz stimulus (alone?).

Figure 7C shows results from clicks and pure-tones up to 32 kHz. The y-axis in 7C is labelled as dB, while in the Methods it is indicated that the units are dB SPL.

According to the Methods the pure-tone stimuli were altered in 10 dB steps. Given that paradigm it is surprising that a statistically significant difference of 10 dB could be found between controls and TG mice at multiple stimulus levels. The error bars on the Figure are not indicated in the legend - are they stdev or SE? In either case they are surprisingly small given 10 dB steps and the small numbers of tested animals.

When attempting to demonstrate subtle threshold differences it is important that the ABR scoring be blinded (was it?) and that example traces from the two genotypes be shown.

It is also important that the data for males and females be presented independently - the authors state that there was no difference but they should show the data. On a C57Bl6 background there are often greater than 5-10 dB threshold differences between male and female WT littermates, although more reliably at ages older than 25 weeks.

Thank you for the careful review and thoughtful suggestions.

1) Regarding the method of ABR analysis, 40,000 Hz was not the stimulus rate, but the sampling rate. We have revised the description as follows: "ABR waveforms using sound stimuli of clicks or tone bursts at 8 kHz, 16 kHz, 24 kHz or 32 kHz (10 kHz, 20 kHz, or 40 kHz for Dia1-KO mice) were recorded for 12.8 ms at a sampling rate of 40,000 Hz by using 50–5000 Hz band-pass filter settings, and ABR waveforms from 500 stimuli were averaged."

2) Thank you for pointing out this error. We have corrected the unit of all ABR data into dB SPL.

3) As described in the statistical analysis section of "Materials and Methods", all data including ABR data are shown as the mean \pm SE (SEM). To clearly show this, we added this description in figure legends of Figure 6C, Figure 8A, Fig. EV3B and Fig. EV4A. In addition, to more clearly show error bars in ABR we have changed from thick to thin lines in Figure 6C, Figure 8A, Fig. EV3B and Fig. EV4A.

4) During the process of revision, we rigorously determined ABR thresholds. Two more otologists in addition to the primary examiner were included to inspect the data; both of them determined the ABR thresholds from the data in blind. The data were not deemed significantly different by any of the three examiners, and we have included a new Figure 8A using the averaged thresholds of the three sets of data. The difference in SE between the previous data and the new one was within the range of error, which supports the statistical significance detected in our data. The description of the methods of ABR analysis is also revised corresponding to the addition described above: page 23. Also, example traces (click and 24 kHz stimulation) from control and TG mice are shown in a new Fig. EV3A.

5) In response to the reviewer's request, we analyzed the ABR data for males and females separately at the age of 25 weeks. The final n of control male and TG male is 18 and 24, respectively; the final n of control female and TG female is 15 and 18, respectively. In this separate analysis, ABR data from male and female were not significantly different, as shown in the new Fig. EV3B. According to these modifications, we replaced ABR data including both male and female at the age of 25 weeks (right end of the Figure 8A; control: n = 33 (male = 18 + female = 15) and DIA1(R1204X)-TG: n = 42 (male = 24 + female = 18), which additionally show the statistical significance at 32 kHz. In addition, we rewrote the animal section of "Materials and Methods" describing the sex of the animals used (page 20).

9. The authors show representative SEM images comparing the cochleae of control and TG animals in 7D, and the OHC loss depicted for the TG genotype is provocative. However, OHC on a C57Bl6 background is common and dependent on age and position in the cochlea. Could the authors provide more quantitative data? e.g. Hair cell counts per length-window at base, middle and apex? Given the subtle ABR differences it would be surprising that there would be a pronounced histological difference that need not be quantified. Finally, if the main difference is in OHC numbers could the authors show DPOAE results as well?

As suggested by the reviewer, HC (both OHC and IHC) loss in TG and control mice at the age of 25 weeks was quantitatively analysed at three portions of cochlea (n = 5). The results are shown in

the new Figure 8C. We also examined DPOAE of 25-week-old TG and control mice, and detected significant differences (control: n=30, TG: n=37). The result is shown in the new Figure 8D.

2nd Editorial Decision

06 September 2016

Thank you for the submission of your revised manuscript to EMBO Molecular Medicine. We have now received the enclosed report from the referee who was asked to re-assess it. As you will see this reviewer is now supportive and I am pleased to inform you that we will be able to accept your manuscript pending final amendments.

Please submit your revised manuscript within two weeks. I look forward to seeing a revised form of your manuscript as soon as possible.

***** Reviewer's comments *****

Referee #2 (Remarks):

My concerns regarding the ABR testing and depiction of the results have all been adequately addressed in the paper. The paper is greatly improved by the inclusion of the DIA1-KO data and careful consideration of the possibility of nonsense-mediated decay. The revised wording also clarifies the misunderstanding I had about the mouse versus human forms of DIA1 in the elongation assay. My concerns regarding the ABR testing and depiction of the results have all been adequately addressed in the paper. This is an excellent paper.

Corresponding Author Name: Takehiko Ueyama

Manuscript Number: EMM-2016-06609